# Genome-Wide Identification of the *MAPK* and *MAPKK* Gene Families in Response to Cold Stress in *Prunus mume*

**DOI:** 10.3390/ijms24108829

**Published:** 2023-05-16

**Authors:** Zhenying Wen, Mingyu Li, Juan Meng, Runtian Miao, Xu Liu, Dongqing Fan, Wenjuan Lv, Tangren Cheng, Qixiang Zhang, Lidan Sun

**Affiliations:** 1Beijing Key Laboratory of Ornamental Plants Germplasm Innovation and Molecular Breeding, National Engineering Research Center for Floriculture, Beijing Laboratory of Urban and Rural Ecological Environment, School of Landscape Architecture, Beijing Forestry University, Beijing 100083, China; 2Center for Computational Biology, College of Biological Sciences and Technology, Beijing Forestry University, Beijing 100083, China

**Keywords:** *Prunus mume*, *MPK* and *MKK* gene families, gene expression, cold response

## Abstract

Protein kinases of the MAPK cascade family (MAPKKK–MAPKK–MAPK) play an essential role in plant stress response and hormone signal transduction. However, their role in the cold hardiness of *Prunus mume* (Mei), a class of ornamental woody plant, remains unclear. In this study, we use bioinformatic approaches to assess and analyze two related protein kinase families, namely, MAP kinases (MPKs) and MAPK kinases (MKKs), in wild *P. mume* and its variety *P. mume* var. *tortuosa*. We identify 11 *PmMPK* and 7 *PmMKK* genes in the former species and 12 *PmvMPK* and 7 *PmvMKK* genes in the latter species, and we investigate whether and how these gene families contribute to cold stress responses. Members of the *MPK* and *MKK* gene families located on seven and four chromosomes of both species are free of tandem duplication. Four, three, and one segment duplication events are exhibited in *PmMPK*, *PmvMPK,* and *PmMKK*, respectively, suggesting that segment duplications play an essential role in the expansion and evolution of *P. mume* and its gene variety. Moreover, synteny analysis suggests that most *MPK* and *MKK* genes have similar origins and involved similar evolutionary processes in *P. mume* and its variety. A cis-acting regulatory element analysis shows that *MPK* and *MKK* genes may function in *P. mume* and its variety’s development, modulating processes such as light response, anaerobic induction, and abscisic acid response as well as responses to a variety of stresses, such as low temperature and drought. Most *PmMPKs* and *PmMKKs* exhibited tissue-specifific expression patterns, as well as time-specific expression patterns that protect them through cold. In a low-temperature treatment experiment with the cold-tolerant cultivar *P. mume* ‘Songchun’ and the cold-sensitive cultivar ‘Lve’, we find that almost all *PmMPK* and *PmMKK* genes, especially *PmMPK3/5/6/20* and *PmMKK2/3/6*, dramatically respond to cold stress as treatment duration increases. This study introduces the possibility that these family members contribute to *P. mume*’s cold stress response. Further investigation is warranted to understand the mechanistic functions of MAPK and MAPKK proteins in *P. mume* development and response to cold stress.

## 1. Introduction

During growth and development, plants are continuously stimulated by mixed signals from internal and external environments. To adapt to changes in the external environment, they have gradually formed complex and delicate signal transduction mechanisms [1,2,3,4,5,6,7,8,9]. The mitogen-activated protein kinase (MAPK or MPKs) cascade pathway is a highly conserved signaling module in eukaryotes that links different extracellular stimuli to a broad range of intracellular responses through a cascade of phosphorylation reactions, which are thought to play vital roles in a variety of biological processes [10,11,12]. The MAPK cascade contains three core gene families: mitogen-active protein kinase kinase kinase (MAPKKK, MAP3K, or MEKK), mitogen-active protein kinase kinase (MAPKK, MAP2K, MKK, or MEK), and the mitogen-active protein kinase (MAPK or MPK) gene family [13]. The MAPK signaling mechanism sequentially phosphorylates and activates each component of the MAPK cascade pathway to transmit and amplify the signal [14].

MAPKKKs are situated in the foremost upstream region of the MAPK cascade pathway and are activated by the phosphorylation of the receptor proteins that receive the signal upstream. Activated MAPKKK conveys the signal by phosphorylating downstream MAPKK through phosphorylating a serine/threonine residue in the activated loop S/T-xxxxx-S/T (S: serine; X: any amino acid; T: threonine) motif [15]. MAPKKs are located at the center of the MAPK cascade, and the activated MAPKK similarly phosphorylates and activates downstream MAPK by phosphorylating tyrosine/threonine residues in TXY (T: threonine; X: any amino acid; Y: tyrosine) motifs in the MAPK activation loop [16]. MAPKs are located downstream of the MAPK cascade, and activated MAPK remain in the cytoplasm to activate other downstream proteins, such as enzymes and cytoskeleton proteins, or enter the nucleus to regulate gene expression by activating transcription factors [16,17].

Multiple domains separate MAPKs from other types of proteins. Towards the N-terminus, which precedes the activation domain, there are eight domains named I–V, VIa–VIb, and VII, and at the end of the activation domain, there are typically activation motifs (T-X-Y) of MAPKs. Towards the C-terminal end of the activation domain, there are five other domains: VIII, IX, X, XI, and CD [18]. There are 20 *MAPKs* identified in Arabidopsis. Four groups of Arabidopsis *MAPK* genes—designated A, B, C, and D—can be identified through phylogenetic analysis [14]. The activation loop of groups A, B, and C contains a -TEY- motif, whereas that of group D contains a -TDY- motif [19]. The eight specific domains of the MAPKK proteins, I through VIII, as well as the ATP-binding, MAPK-binding, activation, and NTF2-binding domains, are highly conserved and contribute to the high similarity of the MAPKK proteins. Ten *MAPKKs* have been identified in Arabidopsis. Based on phylogenetic analysis, *MAPKK* genes in Arabidopsis can also be classified into four groups: A, B, C, and D [14]. NTF2-binding domains only exist in group B (*MKK3*). The phylogenetic relationships of the *MAPK* and *MAPKK* genes described below are based on Arabidopsis results. The terms *MPKs* and *MKKs* are used here as members of specific families of *MAPK* and *MAPKK* genes.

Advances in whole-genome sequencing have made it possible to identify *MAPK* cascade genes across the genome in many different species, such as *Oryza sativa* [20], *Populus trichocarpa* [21], *Helianthus annuus* [22], *Solanum lycopersicum* [23,24], *Glycine max* [25], *Solanum tuberosum, Solanum melongena, Capsicum annuum, Coffea canephora* [24], *Vitis vinfera* [26], *Cucumis sativus* [27], *Brachypodium distachyon* [28], *Musa acuminata* [29], *Jatropha curcas* [19], *Camellia sinensis* [30], and cultivated strawberry [31]. It has been demonstrated that the *MAPK* cascade is crucial for plant development as well as various stress and signaling substance responses [32]. In analyzing the *MAPK* cascade gene family in the model plant Arabidopsis, it is found that the *MEKK1*-*MKK4/5*-*MPK3/6* module participates in the flagellin-triggered immune response [33], whereas the module *MKK4/5*-*MPK3/6* may participate in the regulation of root and embryo development [34,35]. The *GhMAP3K14*-*GhMKK11*-*GhMPK31* module is involved in the drought response in cotton [36]. The *MEKK1*-*MEK1/2*-*MPK4/6* module is activated in response to various forms of stress and has contributed to freezing tolerance in Arabidopsis [37,38,39,40]. In addition, studies have also shown that *ClMAPK7* in watermelon (*Citrullus lanatus*) can react to stresses such as salt, heat, drought, and low temperature [41]. The expression of *MAPKs* increased significantly under cold treatment in *Jatropha curcas* [19]. *ZmMKK4*, a novel group C *MAPKK* in maize (*Zea mays*), confers salt and cold tolerance in transgenic Arabidopsis [42]. *MusaMPK5* is associated with regulating cold tolerance in bananas [43]. *LeMAPK4* participates in cold-induced ethylene production in tomato fruit [44].

In summary, the first step in the initial prediction of gene function in controlling plant responses is to identify *MAPK* cascade genes in plants. However, no comprehensive genomic analysis of the *MAPK* cascade family in *Prunus mume* (also known as Mei) has been conducted. The traditional flower *P. mume* is a collective name for a class of plants indigenous to southwest China as well as the middle and lower reaches of the Yangtze River. This species harbors ornamental value, including fragrance, wonderful flower types, and rich colors. There are now more than 300 known varieties, and complete genome sequencing has been completed for wild *P. mume* and its variety *P. mume* var. *tortuosa*. The wild *P. mume* and *P. mume* var. *tortuosa* all have typical mei twigs and leaves, twigs green or with green ground colours, flowers typical with real mei flower fragrance. The petals of the wild *P. mume* are usually single, with a petal number of five, while those of *P. mume* var. *tortuosa* are multi-layered, usually four-layered, and cream in colour. The most important feature of *P. mume* var. *tortuosa*, in contrast to the wild *P. mume*, is the scattered and indeterminate crown, resembling a swimming dragon, with branches and twigs naturally contorted, while the wild *P. mume* branches and twigs not naturally contorted (Figure 1). At the same time, *P. mume* var. *tortuosa* is also the only variety of *P. mume* to have tortuous branches, and is of particular ornamental value due to its naturally tortuous branches and attractive flowers, which play an important role in urban landscaping. However, low temperatures in northern China severely restrict the species’ ability to grow and expand. Although *MPKs* and *MKKs* have been linked to response to cold stress in other species, little is known about their function in *P. mume*. The completion of a growing number of plant whole-genome sequences has thus far allowed us to identify members of the MAPK cascade family in *P. mume* and its variety *P. mume* var. *tortuosa* as well as in related Rosaceae species. The goal of this study is to perform a genome-wide analysis of the *MPK* and *MKK* gene families in *P. mume* and its variety *P. mume* var. *tortuosa* along with five other closely related Rosaceae species. Based on the sequence comparison results, we found that the sequence similarity between the *MPK* and *MKK* genes in wild *P. mume* and *P. mume* var. *tortuosa* was very high. Therefore, the expression of the *PmMPK* and *PmMKK* genes in wild *P. mume* was examined using the transcriptome data obtained based on wild *P. mume* as a reference genome analysis. In addition, to validate the results of the transcriptome data, we also examined the response of the *PmMPK* and *PmMKK* genes to cold stress in *P. mume* using a real-time fluorescence quantitative assay. This analysis serves as a starting point for a more in-depth investigation of the potential functional roles of members of the *MPK* and *MKK* gene families in *P. mume*, its variety *P. mume* var. *tortuosa* and Rosaceae.

## 2. Results

### 2.1. Identification of MPK and MKK Gene Family Members

A total of 11, 12, and 69 *MPKs* were identified in *P. mume*, *P. mume* var. *tortuosa* genome, and the other five Rosaceae species (including 12 in *P. armeniaca*, 10 in *P. persica*, 13 in *P. salicina*, 22 in *M. domestica*, and 12 in *R. chinensis*), respectively. Furthermore, 7 *MKKs* were detected in *P. mume*, 7 were detected in *P. mume* var. *tortuosa* genome, and 36 were detected in the other five Rosaceae species (including 7 in *P. armeniaca*, 9 in *P. persica*, 5 in *P. salicina*, 8 in *M. domestica*, and 7 in *R. chinensis*) (sequence details are shown in File S1). Each newly detected *MPK* and *MKK* gene was given a name based on its similarity to Arabidopsis MPK and MKK proteins (Table 1, Appendix A). The number of amino acids, molecular weight (MW), and isoelectric point (pI) were computed based on the identified MPK and MKK proteins’ sequences. As shown in Table 1, the predicted *Pm*MPK proteins in *P. mume* ranged in length from 368 (*PmMPK7*) to 818 (*PmMPK8*) amino acids with relative molecular weights of 42.39 kDa (*PmMPK7*) to 92.92 kDa (*PmMPK8*) and theoretical pIs of 5.18 (*PmMPK13*) to 9.31 (*PmMPK19*). Moreover, 7 *PmMKKs* were predicted to encode 324–518 aa, with MW ranging from 36.17–57.79 kDa and pIs from 5.36–8.04. In *P. mume* var. *tortuosa*, 12 *PmvMPKs* were predicted to encode 370–1011 aa, with MW ranging from 42.58–115.17 kDa and pIs from 5.18–9.29. Furthermore, 7 *PmvMKKs* were predicted to encode 321–518 aa, with MW ranging from 35.95–57.77 kDa and pIs from 5.51–7.58. It can be seen that the number of amino acids, molecular weight, and isoelectric point of *P. mume* var. *tortuosa* are relatively similar to those of wild *P. mume*, and only one gene, *PmvMPK7*, had several indices higher than the others. Subcellular localization was predicted, showing that *MPKs* and *MKKs* in *P. mume* and *P. mume* var. *tortuosa* are located in the nucleus with the exception of *PmvMPK7*, which may be present in the cell membrane.

### 2.2. MPK and MKK Genes’ Phylogenetic Analysis and Classification

To comprehend how homologous *MPK* and *MKK* genes have evolved, we established phylogenetic trees of all *MPK* and *MKK* sequences of *A. thaliana* (model dicots), *O. sativa* (model monocots), *P. mume,* and *P. mume* var. *tortuosa* using the ML method. According to *AtMPK* and *AtMKK* reported in earlier studies [14], *MPK* and *MKK* in *P. mume* and *P. mume* var. *tortuosa* are classified into four clades (i.e., Clade A, B, C, and D) (Appendix A). In order to examine the evolutionary relationship between *MPK* and *MKK* in *P. mume* and *P. mume* var. *tortuosa* and Rosaceae species, an ML phylogeny was constructed using nine species, including five other Rosaceae species. The *MPK* and *MKK* gene family members from the nine species were divided into four clades (Figure 2A). Clade D had the highest number of the *MPK* gene, comprising 4 *PmMPKs*, 5 *PmvMPKs*, 8 *AtMPKs*, 11 *OsMPKs*, and 27 Rosaceae *MPKs*. The number of *MPK* genes found in the evolutionary clades of A, B, and C did not differ significantly. Clades A, B, and C had two, three, and two *PmMPKs* and *PmvMPKs*, respectively (Figure 2A and Appendix A). Clade D also had the highest numbers of the *MKK* gene, comprising 4 *PmMKKs*, 4 *PmvMKKs*, 4 *AtMKKs*, 3 *OsMKKs*, and 19 Rosaceae *MKKs*. Notably, Clade C had only four members, i.e., two *AtMKKs* and two *OsMKKs*. None of the selected species in Rosaceae were found in Clade C. Each of the nine species had one member in Clade B. Clade A had a similar number of members, ranging from two to four; for example, there were two *PmMKKs* and two *PmvMKKs* (Figure 2B; the exact number is shown in Appendix A). It can be seen that in the *MPK* and *MKK* families, the number of genes in each taxonomic clade is equal for both wild *P. mume* and *P. mume* var. *tortuosa*, except for the gene *PmvMPK7*, which was more abundant in *P. mume* var. *tortuosa* than wild *P. mume*.

### 2.3. Conserved Motif, Domain, and Gene Structure of MPK and MKK Proteins in *P. mume* and *P. mume* var. tortuosa

To investigate the sequence features of MPK and MKK proteins in *P. mume* and *P. mume* var. *tortuosa*, the MEME program and TBtools were utilized to predict and map conserved domains. A total of 13 unique motifs were investigated in MPK and MKK proteins, which are illustrated schematically in Figure 3. The number of motifs in the MPKs was similar, ranging from 9 to 10. Motifs 1, 2, 3, 4, 5, 6, and 12 were highly conserved and existed in all 11 *Pm*MPK and 12 *Pmv*MPK proteins. Motifs 7 and 8 were present only in Clade D. Motifs 9 and 10 were present only in Clades A, B, and C. Motif 11 was present only in Clade A and B; motif 13 existed only in four members (*PmMPK19*, *PmvMPK19*, *PmMPK20*, *PmvMPK20*) of Clade D (Figure 3). The number of MKK motifs in *P. mume* and *P. mume* var. *tortuosa* ranged from 7 to 10. Motifs 1, 4, and 5 were highly conserved and were present in all 7 *Pm*MKK and 7 *Pmv*MKK proteins. Motif 3 was found in 13 MKK proteins in *P. mume* and *P. mume* var. *tortuosa* with the exception of *Pm*MKK2. Motif 6 was present in 12 MKK proteins in *P. mume* and *P. mume* var. *tortuosa* with the exception of *Pm*MKK3 and *Pmv*MKK3. Motifs 7, 9, and 10 were present only in Clades A and B. Motifs 11, 12, and 13 were present only in Clade B (Figure 4A). Detailed information on MPK and MKK motifs is shown with logos acquired from the MEME Suite website in Appendix A. It can be seen that the conserved motif composition and gene structure of the homologous genes of wild *P. mume* and *P. mume* var. *tortuosa* were identical.

The multiple sequence alignment of MPKs in *P. mume* and *P. mume* var. *tortuosa* showed that they all contained the characterized multiple domains I-V, VIa, VIb, VII, VIII, IX, X, XI, and CD-domain (Figure 5) [18]; Clades A, B, and C contained TEY activation motifs, while Clade D contained TDY activation motifs (Appendix A). Multiple alignments of the MKK sequences in *P. mume* and *P. mume* var. *tortuosa* were generated, showing that MKK proteins are highly similar and that the eight specific domains, as well as ATP-binding, MAPK-binding, and activation domains, are highly conserved. By contrast, the NTF2-binding domains only existed in Clade B (*Pm*MKK3 and *Pmv*MKK3) (Figure 6 and S6). In addition, we conducted a comparative analysis of the homologous genes of the *MPK* and *MKK* gene families in *P. mume* and *P. mume* var. *tortuosa*, respectively, and found that sequence similarity for homologous gene pairs was above 93% in all cases except *Pm*MPK7*/Pmv*MPK7, as shown in Appendix A. Sequence alignment shows that *Pmv*MPK7 had 643 bp more sequence than *Pm*MPK7, which was not a structural domain of the *MPK* gene family.

In order to clarify the structural characteristics of the *MPKs* and *MKKs* in *P. mume* and *P. mume* var. *tortuosa*, the exon–intron structure was examined further. As shown in Figure 3B, the number of introns in the *MPK* gene family members varied widely from 1 to 15; those in Clade A and Clade B each had 4–5 introns. Clade C family members had one intron with the exception of *PmvMPK7*, which contained 15 introns, which should be related to the fact that it had an additional part of the sequence than its homolog *PmMPK7*. There were 8–12 introns in Clade D family members. Similarly, *P. mume* and *P. mume* var. *tortuosa*’s *MKK* gene family members demonstrated a wide range of intron counts from 0 to 8, with those in Clade D having no introns. Those in Clade B contained eight introns; those in Clade A contained seven, excluding *PmMKK6*, which contained eight (Figure 4B).

### 2.4. Chromosomal Distribution and Gene Duplication Analysis

All the *MPKs* and *MKKs* in *P. mume* and *P. mume* var. *tortuosa* were mapped based on gene location information, which showed that they were all located on the chromosome. The distribution of *MPK* genes on the chromosome was similar in *P. mume* and *P. mume* var. *tortuosa*. Chromosomes 1, 2, 7, and 8 each contained 2 *MPKs*, whereas chromosomes 3, 4, and 5 each contained 1 (Figure 7). Likewise, the distribution of *MKK* genes on chromosomes was also similar in *P. mume* and *P. mume* var. *tortuosa*, with chromosomes 2, 7, and 8 each containing two *MKKs* and chromosome 4 containing one (Figure 7). The chromosome distributions of the other five species of Rosaceae are given in Appendix A.

Tandem and segmental duplication are the two main engines for generating new copies of genes in the evolution of gene families. Tandem duplication generates nearby copies of genes in genomic clusters, and segmental duplication events have a different effect, as they may widely disperse copies of genes throughout the genome [45]. We use Multiple Collinearity Scan Toolkit (MCScanX) and Advanced Circos in TBtools to analyze gene tandem and segment replication events. The results showed that no tandem duplication events were detected in *P. mume*, *P. mume* var. *tortuosa*, or any of the other five Rosaceae species.

A synteny analysis of *MPKs* and *MKKs* in *P. mume* and *P. mume* var. *tortuosa* was conducted using the Advanced Circos procedure of TBtools. Four segmental duplication events, including *PmMPK13*/*PmMPK20*, *PmMPK19*/*PmMPK20*, *PmMPK3*/*PmMPK5*, and *PmMPK1*/*PmMPK7*, were identified in the *MPK* gene family of *P. mume*, and three segmental duplication events, including *PmvMPK3*/*PmvMPK12*, *PmvMPK4*/*PmvMPK12*, and *PmvMPK1*/*PmvMPK7*, were detected in *P. mume* var. *tortuosa*. In addition, only one segmental duplication event was identified in the *MKK* gene family of *P. mume*, namely *PmMKK9-3*/*PmMKK3*. Each collinear gene pair was situated on a different chromosome, as shown by the red, blue, and olive-drab lines in Figure 8A. No *MKK* gene segment duplication events were detected in *P. mume* var. *tortuosa*. In addition, we analyzed segmental duplication events in 5 other Rosaceae species, of which 2 were detected in the *MPK* gene family of *P. armeniaca*, 9 in *P. persica*, 5 in *P. salicina*, 24 in *M. domestica*, and 2 in *R. chinensis*. It is worth noting that there were two *MKK* genes in the family of *P. persica*, one in *P. salicina*, and two in *M. domestica*. No *MKK* gene segment duplication events were detected in *P. armeniaca* and *R. chinensis*, the details of which are given in Appendix A.

In order to detect the selection pressure during gene divergence after duplication, the Ka (nonsynonymous)/Ks (synonymous) substitution ratio and divergence time of the duplicated pairs were further calculated. In the evolutionary process, the Ka/Ks ratio > 1 indicates positive selection (adaptive evolution), a ratio = 1 indicates neutral evolution (drift), and a ratio < 1 indicates negative selection (conservation). Result showed that the Ka/Ks ratios for all the duplicated orthologous gene pairs were all <1, indicating that *MPK* and *MKK* genes in *P. mume*, *P. mume* var. *tortuosa*, and the five Rosaceae species primarily undergo purifying selection following their duplication. Appendix A illustrates the divergence times of the duplicated gene pairs. The Ka/Ks ratio could not be calculated for some of the duplicated gene pairs, which was possibly because these gene pairs exhibited synonymous mutation at sites where synonymous mutations could occur; that is, the sequence divergence was large, resulting in a large evolutionary distance.

### 2.5. Interspecies Collinearity Analysis of the MPK and MKK Gene Family

In order to further investigate the particular retention of *MPKs* and *MKKs* in *P. mume* and *P. mume* var. *tortuosa*, the collinear relationships of these species with *A. thaliana* and five Rosaceae species were analyzed using the MCScanX algorithm of TBtools. A total of 15 homologous *MPK* gene pairs were found in *P. mume* and *P. mume* var. *tortuosa* (Figure 8B, red lines). Furthermore, it was found that *P. mume* and *A. thaliana* share 16 pairs of homologous genes, as do *P. mume* and *P. armeniaca*. In addition, 14 pairs are shared between *P. mume* and *P. persica*; 17 between *P. mume* and *P. salicina*; 29 between *P. mume* and *M. domestica*; and 14 between *P. mume* and *R. chinensis* (Figure 9A–C and Appendix A). Similarly, 20 homologous gene pairs were detected between *P. mume* var. *tortuosa* and *A. thaliana*; 18 between *P. mume* var. *tortuosa* and *P. armeniaca*; 14 between *P. mume* var. *tortuosa* and *P. persica*; 17 between *P. mume* var. *tortuosa* and *P. salicina*; 29 between *P. mume* var. *tortuosa* and *M. domestica*; and 17 between *P. mume* var. *tortuosa* and *R. chinensis* (Figure 9D–F and Appendix A).

A total of seven pairs of homologous *MKK* genes were found to be shared between *P. mume* and *P. mume* var. *tortuosa* (Figure 7B, blue lines). Similarly, it was found that *P. mume* and *A. thaliana* share five pairs of the homologous genes; *P. mume* and *P. armeniaca* share seven; *P. mume* and *P. persica* share seven; *P. mume* and *P. salicina* share five; *P. mume* and *M. domestica* share seven; and *P. mume* and *R. chinensis* share five (Figure 10A–C and Appendix A). Furthermore, six pairs of homologous genes were found to be shared by *P. mume* var. *tortuosa* and *A. thaliana*; seven are shared by *P. mume* var. *tortuosa* and *P. armeniaca*; seven are shared by *P. mume* var. *tortuosa* and *P. persica*; five are shared by *P. mume* var. *tortuosa* and *P. salicina*; six are shared by *P. mume* var. *tortuosa* and *M. domestica*; and five are shared by *P. mume* var. *tortuosa* and *R. chinensis* (Figure 10D–F and Appendix A).

### 2.6. Cis-Acting Elements in MPK and MKK Gene Promoter Prediction Analysis

To further examine the potential regulatory mechanisms of *MPKs* and *MKKs* during *P. mume* and *P. mume* var. *tortuosa* growth or defense mechanisms, especially in response to abiotic stresses such as low temperature, we uploaded the 2.0 kb sequence upstream of each *MPK* and *MKK* gene translation start point to the PlantCARE database to search for specific cis-acting elements. Fourteen and thirteen conserved regulatory elements related to plant hormones and environmental stress responses were analyzed in the *MPK* and *MKK* promoters, respectively (Figure 11 and Appendix A, and Appendix A and Appendix A). The promoter regions of *P. mume* and *P. mume* var. *tortuosa* toward the *MPK/MKK* gene family were found to contain several elements related to light response, anaerobic induction, MeJA response, and ABA response. Based on the regulatory elements in their promoters, six *MPK* gene family members in *P. mume* and *P. mume* var. *tortuosa* are sensitive to low temperature. Similarly, five *MKK* gene family members are sensitive to low temperatures in *P. mume* and *P. mume* var. *tortuosa*, as shown in Figure 11 and Appendix A and Appendix A and Appendix A. Figure 12 illustrates the number of each cis-element of the *PmMPK*, *PmMKK*, *PmvMPK,* and *PmvMKK* genes. The results showed that they contain comparable numbers of low-temperature cis-elements.

### 2.7. Expression Pattern Analysis of MPKs and MKKs in *P. mume*

Based on the results of the sequence comparison, we found that the sequence similarity between the *MPK* and *MKK* genes in wild *P. mume* and *P. mume* var. *tortuosa* is very high. We therefore examined the expression of the *PmMPK* and *PmMKK* genes in wild *P. mume*. Our RNA-seq dataset was used to analyze the expression patterns of *MPK* and *MKK* family members in the roots, stems, leaves, flower buds, fruits of wild *P. mume* [46], and flower buds of cultivar *P. mume* ‘Lve’ at different stages of dormancy [47] to learn more about the function of *PmMPKs* and *PmMKKs* in development and response to low temperature. Appendix A present their RPKM values. As illustrated in Figure 13A, 11 *PmMPKs* were expressed in all the test tissues, and all showed a high level of transcript accumulation (RPKM > 5), of which two *PmMPK* genes in flower buds (*PmMPK13* and *PmMPK*7), *PmMPK4* in leaves, three *PmMPK* genes in roots (*PmMPK3, PmMPK5* and *PmMPK16*), three *PmMPK* genes in fruits (*PmMPK7, PmMPK8,* and *PmMPK19*), and at least four *PmMPK* genes in stems (*PmMPK1, PmMPK4, PmMPK6,* and *PmMPK20*) have relatively high expression levels. In addition, three (*PmMKK9*–*2*, *PmMKK9*–*3*, and *PmMKK10*) out of seven *PmMKKs* had very low or no detectable expression in the tissues tested (Figure 13A). The remaining four *PmMKKs* were relatively highly expressed in different tissues. For instance, *PmMKK2* and *PmMKK6* levels were high in leaves, *PmMKK9*–*1* and *PmMKK2* levels were high in fruits, and *PmMKK3* levels were high in stems.

All the *PmMPKs* were expressed during the flower bud dormancy period and at particular development stages (Figure 13B). Four *PmMPKs* (*PmMPK1*, *PmMPK3*, *PmMPK7*, and *PmMPK13*) were preferentially expressed during the endo-dormancy I stage (EDI) in November, six *PmMPKs* (*PmMPK3*, *PmMPK5*, *PmMPK7*, *PmMPK16*, *PmMPK19*, and *PmMPK20*) showed the highest expression levels during the endo-dormancy II stage (EDII) in December, and four *PmMPKs* (*PmMPK7*, *PmMPK16*, *PmMPK19*, and *PmMPK20*) showed up-regulated expression during the endo-dormancy III stage (EDIII) in January. Three *PmMPKs* (*PmMPK4*, *PmMPK6*, *PmMPK8*) in particular showed greater expressions during the natural flush stage (NF) in February. Six up-regulated genes (*PmMPK4*, *PmMPK6*, *PmMPK7*, *PmMPK8*, *PmMPK16*, and *PmMPK20*) (Appendix A) exhibited low-temperature response elements in their putative promoter regions. In contrast to three *PmMKKs* (*PmMKK9*–*2*, *PmMKK9*–*3*, and *PmMKK10*) that had very low or no detectable expression in the tissues tested, the other four *PmMKKs* were highly expressed at one or several stages (Figure 13B). For example, *PmMKK2*/*6*/*9*–*1* was preferentially expressed in November in EDI, *PmMKK2*/*6* showed the highest level of expression in EDII, and *PmMKK3*/*6* showed up-regulated expression in EDIII. Among them, *PmMKK2* and *PmMKK6* exhibited low-temperature response elements in their putative promoter regions.

To further evaluate the expression patterns of *PmMPKs* and *PmMKKs* under exposure to cold, we examined the stems of the cold-tolerant cultivar *P. mume* ‘Songchun’ in three different geographic locations, whose FPKM values are provided in Appendix A. As shown in Figure 14A, five *PmMPKs* (*PmMPK3*, *PmMPK7, PmMPK16*, *PmMPK19*, and *PmMPK20*) showed higher expression in winter, of which three *PmMPKs* (*PmMPK3*, *PmMPK16*, and *PmMPK20*) were also highly expressed in autumn in Beijing and Chifeng. Additionally, two *PmMPKs* (*PmMPK1* and *PmMPK6*) showed higher expression levels in winter in Beijing (−5.4 °C). Five *PmMPKs* (*PmMPK1*, *PmMPK4*, *PmMPK7*, *PmMPK13*, and *PmMPK19*) showed higher expression in autumn (6.1~7.9 °C). The expression of *PmMPK8* was higher in the spring (3.2~5.3 °C), and *PmMPK5* showed higher expression levels in the spring of Chifeng and Gongzhuling (3.2, 5.3 °C). Among these genes with up-regulated expression, six *PmMPKs* (*PmMPK4*, *PmMPK6*, *PmMPK7*, *PmMPK8*, *PmMPK16*, and *PmMPK20*) (Appendix A) were found to contain low-temperature response elements in their putative promoter regions. Another heatmap was created to compare the expression patterns of *PmMPKs* for the year (Figure 14B). *PmMPKs* in the samples obtained from the three locations displayed comparable expression patterns at the same times throughout the year, as seen in Figure 14B. Most *PmMPKs* were expressed at higher levels in autumn and winter and down-regulated in the spring (Figure 14B). Similarly, three *PmMKKs* (*PmMKK9*–*2*, *PmMKK9*–*3*, and *PmMKK10*) also had very low or no detectable expression (Figure 14A,B). Of the other four genes, *PmMKK3* was relatively highly expressed in spring, *PmMKK6* was relatively highly expressed in fall in Gongzhuling, *PmMKK2* was relatively highly expressed in winter in Gongzhuling, and *PmMKK9*–*1* was irregularly expressed, with high expression in spring, autumn, and winter in different locations. Seasonal expression patterns for the four genes at the three locations were not obvious.

### 2.8. PmMPK and PmMKK Expression Patterns in Response to Cold Treatment

To validate the results of transcriptome analysis of *P. mume MPK* and *MKK* genes during naturally low-temperature, and evaluate the involvement of *PmMPKs* and *PmMKKs* in cold stress, expression profiles under 4 °C stress treatments for 0, 1, 4, 6, 12, 24, 48, and 72 h were studied using qRT–PCR with the cold-tolerant cultivar *P. mume* ‘Songchun’ and the cold-sensitive cultivar *P. mume* ‘Lve’. We used a qRT–PCR assay on the 11 identified *PmMPKs* and 4 *PmMKKs*, and relative expression was calculated with the *PP2A* and *Actin* genes of *P. mume* as the reference genes, respectively (Figure 15 and Appendix A). All the tested *MPK* and *MKK* genes showed potentially induced expression after 1–72 h of cold stress in the stems (Figure 15 and Appendix A). The expression levels of 11 *PmMPK* and 4 *PmMKK* genes changed at different rates in the two cultivars under the imposed cold stress treatment. Particularly noteworthy was the cold-induced expression of *PmMPK3*, which after 1 h in ‘Lve’ and ‘Songchun’ reached its greatest expression levels (13- and 9-fold increases, respectively), which consistent with the transcriptome results, *PmMPK3* expression was upregulated in ‘Lve’ when the temperature decreases in November and December, and in ‘Songchun’ during the winter months. In addition, the responses of the cold-induced genes could sometimes be remarkably quick, and distinct expression changes were discovered at earlier time points. For example, *PmMPK16* genes were remarkably up-regulated after 1 h of cold stress, while *PmMPK1, PmMPK5, PmMPK6, PmMPK7*, *PmMPK8*, *PmMPK19*, *PmMPK20*, *PmMKK3*, and *PmMKK9*–*1* were significantly down-regulated in ‘Songchun’. In ‘Lve’, *PmMPK1* and *PmMPK3* were remarkably up-regulated after 1 h of cold stress, while *PmMPK6, PmMPK13, PmMPK20, PmMKK3, PmMKK6,* and *PmMKK9*–*1* were remarkably down-regulated. The expression trends of these genes in the face of sudden cooling were almost identical to the transcriptome results obtained in the autumn of ‘Songchun’ in Beijing, when the temperature starts to drop, and in the buds of ‘Lve’ in November, when the temperature drops (Figure 13B and Figure 14B). The expression of the four genes *PmMPK5*, *PmMPK6*, *PmMPK20*, and *PmMKK3* decreased in the two cultivars as the length of the treatment increased. The expression patterns of *PmMPK3, PmMPK19,* and *PmMKK9-1* were similar in both varieties, with expressions first up-regulated and then down-regulated or first down-regulated and then up-regulated after cold treatment. The expression patterns of five genes (*PmMPK4*, *PmMPK7, PmMPK8, PmMKK2*, and *PmMKK6*) varied between the two cultivars. Of these five genes, *PmMPK4/7/8* showed a decreasing trend in ‘Songchun’, while in ‘Lve’ it showed a rising trend as the length of the treatment increased. *PmMKK2* was initially up-regulated, followed by down-regulation as the length of the treatment increasedin ‘Songchun’, while in ‘Lve’ it exhibited no significant changes overall. *PmMKK6* was up-regulated with treatment increase in ‘Songchun’, while no significant changes with prolonged treatment were demonstrated in ‘Lve’.

## 3. Discussion

MAP kinase cascades in plants are the oldest conserved signal transduction pathways. They regulate many biological functions, including hormone signaling, growth, and development [12,14,48,49] as well as various types of stress [10,11,12,50]. The complete genome-wide sequencing of a large number of species has led to the identification of numerous *MPK* cascade family members in numerous plants [19,20,21,22,23,24,25,26,27,28,29,30,31]. However, *P. mume* and its variety *P. mume* var. *tortuosa* MAPK cascade genes’ identification and functions remain mostly unknown. *P. mume* is the collective name for an essential early spring flowering plant in China and Southeast Asia that is often challenged by cold stress in northern China. In order to investigate the tolerance mechanisms of *P. mume* and its variety *P. mume* var. *tortuosa* for cold stresses, we conducted a thorough analysis of the *MPK* and *MKK* family genes in wild *P. mume, *P. mume** var. *tortuosa*, and five associated Rosaceae species along with the expression traits of *PmMPK* and *PmMKK* genes under cold stresses. The present study used Arabidopsis protein sequences from 20 *AtMPKs* and 10 *AtMKKs* to identify the complete set of *MPK* and *MKK* proteins in *P. mume*, *P. mume* var. *tortuosa*, and five Rosaceae species. A total of 11 *PmMPKs* and 7 *PmMKKs* from *P. mume*; 12 *PmvMPKs* and 7 *PmvMKKs* from *P. mume* var. *tortuosa*; 12 *PaMPKs* and 7 *PaMKKs* from *P. armeniaca*; 10 *PpMPKs* and 9 *PpMKKs* from *P. persica*; 13 *PsMPKs* and 5 *PsMKKs* from *P. salicina*; 22 *MdMPKs* and 8 *MdMKKs* from *M. domestica*; and 12 *RcMPKs* and 7 *RcMKKs* from *R. chinensis* were identified (Appendix A). The genome sizes of these plants are ~219.9 Mb for *P. mume*, ~237.7 Mb for *P. mume* var. *tortuosa*, ~221.9 Mb for *P. armeniaca*, ~224.6 Mb for *P. persica*, ~284.2 Mb for *P. salicina*, ~658.9 Mb for *M. domestica*, and ~503 Mb for *R. chinensis* [46,51,52,53,54,55,56]. This phenomenon suggests that there is no direct correlation between the number of *MPK* and *MKK* genes and plant genome size. The numbers of *MPK* gene families among the selected Rosaceae species were similar with the exception of the apple, which had 29 *MdMPK* genes—a number that is far greater than those of other species. Such a high number of homologous genes in apple is consistent with the extensive events of duplication in its genome [57].

According to Arabidopsis [14], these MPKs and MKKs are classified into four clades (A–D). The MPK proteins in Clades A, B, and C all have TEY phosphorylation sites, while the protein in D has TDY phosphorylation sites. A close genetic association was found between *P. mume* and *P. mume* var. *tortuosa* in terms of *MPK* and *MKK* genes. Consistent clade division and high sequence similarity were demonstrated (Appendix A), indicating that *MPK* and *MKK* members in *P. mume* and *P. mume* var. *tortuosa* are relatively conservative in evolution, excepting *PmvMPK7*, which is ~640 bp longer than its homologous gene *PmMPK7*. This extra sequence belongs to other genes according to the annotation (Appendix A) and may have been caused by a gene annotation error. In addition, *P. mume* var. *tortuosa* has one additional gene, *PmvMPK17*, compared with wild *P. mume*, which may be related to the expansion of the *MPK* gene family after the divergence of the two species and their independent evolution [51]. The gene numbers of Clades A, B, and C varied little among the different species studied. Clade D was demonstrated to have the largest number of members (Appendix A), which is consistent with previous studies [14,58,59]. It is worth noting that no *MKK* family members from Clade C were found in the selected Rosaceae species, and the numbers from Clade A and B were similar to those of Arabidopsis and rice, indicating that members of Clade C tended to evolve into Clade D during the evolutionary process (Figure 2B). The NTF2-binding domain only existed in Clade B (Figure 6), which is consistent with previous studies on angiosperms, gymnosperms, pteridophytes, and bryophytes [14,18,59].

The structure of gene family members is closely linked to gene expression and function [60]. Members within the same clade have similar motif compositions. Different clades contain 1–2 unique motifs within them, suggesting that the conserved motifs in *MPK* and *MKK* genes support their close evolutionary relationship. Unique motifs may participate in diverse biological processes between different clades [61]. Exon/intron structure is essential to the biological evolution of organisms [62]. In this study, it was found that similar intron–exon structures exist in members belonging to the same clade of *MPK* and *MKK* families, with the exception of *PmvMPK7*, which had 15 introns—far more than members of the same clade and consistent with its long genetic sequence possibly due to a gene annotation error (Appendix A). The *MKK* gene family was known to have a smaller number of gene families and the ability to participate in multiple biological processes. Being intermediate between *MAPK* and *MAPKKK* and able to react with multiple *MAPKs* and *MAPKKKs* to form multiple distinct signaling pathways, *MAPKK* may be the intersection of signaling networks in plants consisting of multiple MAPK cascade pathways [14,21].

Segmental and tandem duplications have been proposed to represent two of the primary causes of gene family expansion in plants [45]. Our data showed that no tandem duplication events occur in the selected Rosaceae species, while segment duplication is frequently present in Rosaceae genomes, especially in the *MPK* gene family (Appendix A), suggesting that chromosomal segments may play an essential role in the expansion of these gene families. Previous studies have also suggested that tandem duplication is rare in expanding the *MAPK*, *MAPKK*, and *MAPKKK* gene families [19,63,64,65,66]. The intergenomic collinearity analysis of *MPK/MKK* genes from the species investigated in this study found the genes to have good homology, providing further evidence that the *MPK* and *MKK* gene families are conserved in angiosperms (Figure 8, Figure 9 and Figure 10), which may imply functional consistency among these homologous genes.

Gene function and regulation are determined mainly by cis-acting regulatory elements in the promoter region [67]. In the course of long-term evolution, plants have evolved intricate mechanisms of gene regulation in response to adverse environmental impacts. The promoters of both *MPKs* and *MKKs* include stress-related cis-acting elements such as abscisic acid response, anaerobic induction, drought inducibility, and low-temperature responsive cis-acting elements in *P. mume* and *P. mume* var. *tortuosa*. Our results showed that a large proportion (≥1/2) of the number of genes contained low-temperature-associated cis-elements in both the *MPK* and *MKK* gene families of *P. mume* and *P. mume* var. *tortuosa*. In addition, the promoters of the two gene families from sugarcane [63] and tea [58] were also found to contain the aforementioned cis-acting elements, and expression levels of *MPKs* and *MKKs* changed after stress treatment.

*MPK* and *MKK* genes play critical roles in a wide range of biological processes [12]. We examined their expression in different tissues of wild *P. mume*, and the results showed preferential tissue expression among *MPK* and *MKK* gene pairs (Figure 13), which supports previous findings [19,59,64]. Duplicated gene pairs of *PmMPK* (*PmMPK20/PmMPK19*; *PmMPK7/PmMPK1*; *PmMPK20/PmMPK13*; and *PmMPK3/PmMPK5*) and *PmMKK* (*PmMKK3/PmMKK9-3*) had different expression patterns. For example, *PmMPK20* was more highly expressed in the stem, but this was not the case for the similar duplicated gene *PmMPK19*. Moreover, the expression of *PmMPK7* was higher in the flower bud and fruit, but its duplication *PmMPK1* was predominantly expressed in the stem and root. This phenomenon has also been observed in *J. curcas* [19]. Thus, regardless of duplicated gene pairs having similar genetic compositions, they may not share similar functions or participate in the same metabolic pathways [27]. In the evolution of *P. mume*, some functions might be lost or gained after duplication.

Several studies have investigated the role of the *MAPK* cascade in response to cold stress in different plants [19,37,38,39,40,41,42,43]. In this study, we found that the *PmMPK* and *PmMKK* genes are specifically expressed during different periods of flower bud dormancy from winter to spring. Thus, we hypothesize that these *PmMPKs* and *PmMKKs* are involved in the cold response of protected flower buds at low temperatures during different stages of flower bud dormancy. In addition, we identified several *PmMPK* genes that are specifically highly expressed in autumn (*PmMPK4*, *PmMPK13*, *PmMPK16*) and winter (−5°~−22°) (*PmMPK1*, *PmMPK3*, *PmMPK6*, *PmMPK7*, *PmMPK14*, *PmMPK19*) (Figure 14). This suggests that the expression of these genes starts to increase in autumn from the onset of cold domestication to withstand the coming cold, continues to increase in winter to help the plant survive the harsh winter, and decreases in spring after temperatures start to rise again. According to promoter analysis, the majority of these genes (5/9) have low-temperature cis-acting elements. By contrast, the other four genes do not have low-temperature response elements (Appendix A) but are induced into expression by low temperatures. Hence, the cis-elements of the genes are not the only determinants of the stress response, which may also be induced by other stresses. This phenomenon has also been observed in other studies [58]. The expression pattern of the *PmMKK* gene at three sites and three periods is less pronounced, suggesting that the *MKK* gene has distinct patterns in stress response across different plant species.

According to the qRT–PCR investigation results, *PmMPKs* and *PmMKKs* were activated at low temperatures (4 °C), and their expression either increased or decreased as the treatment duration was prolonged (Figure 15). The expression trends of many genes in the face of sudden cooling are consistent with the transcriptome results. Some genes (*PmMPK4* and *PmMPK8*) were slightly down-regulated or up-regulated after 1 h of treatment but were significantly up-regulated after 48/72 h of cold stress, indicating the generalized response of these *MAPK* cascade pathway kinase genes to a variety of adversity stresses [42,68]. Four genes (*PmMPK5*, *PmMPK6*, *PmMPK20*, and *PmMKK3*) in ‘Songchun’ and five (*PmMPK5*, *PmMPK6*, *PmMPK13*, *PmMPK20,* and *PmMKK3*) in ‘Lve’ showed a decrease in expression levels with increasing treatment time (Figure 15), suggesting that these genes may be negatively regulated by low temperatures, leading to enhanced cold sensitivity. This phenomenon has also been observed in cucumber and watermelon, where most *CsMAPK* and *ClMAPK* genes are down-regulated in expression after chilling treatment [27,41]. The expression levels of *PmMKK6* in ‘Songchun’ and *PmMPK4* in ‘Lve’ gradually increased with continued treatment (Figure 15), suggesting that these two genes may be positively regulated by cold stress responses and enhance the cold tolerance of *P. mume*. The discrepancies in the expression patterns of *PmMPK4*, *PmMPK7*, *PmMKK8*, *PmMKK2*, and *PmMKK6* between ‘Songchun’ and ‘Lve’ may have been due to genetic variation between these two cultivars that has made them differently resistant to cold. In terms of species classification, ‘Songchun’ belongs to the apricot *mume* and ‘Lve’ to the true *mume*. The branches of Songchun are pale grey bronze violet with pink flowers and pale brown violet sepals, resembling apricots. The branches of ‘Lve’ are pale yellow-green, with pale cream to white flowers and pale yellow-green sepals (Figure 16). Differences in branching morphology between the two cultivars may be one reason why some genes respond differently to cold. Some homologs of the *MAPK* cascade gene showing different expression patterns under the same stress conditions have also been found in other species. For example, *AtMPK7* is remarkably down-regulated after cold stress, while *CsMPK7* is remarkably up-regulated; the *OsMKK4* gene is up-regulated after cold stress, while *CsMKK4* is down-regulated under the same conditions [27]. After cold stress treatment, some *MPK* and *MKK* genes have similar expression patterns, suggesting that these gene pairs may had similar functions. Some homologs are remarkably different, suggesting that they may have acquired new functions in evolution and play roles in different signaling pathways [27,69,70].

## 4. Materials and Methods

### 4.1. Plant Genomic Resources

We downloaded the MPK and MKK proteins from two model plants—*Arabidopsis thaliana* and *Oryza sativa*, which stood in for dicotyledons and monocotyledons, respectively—as well as from the genomic files of five other Rosaceae species to analyze the phylogenetic relationships of *MPK* and *MKK* genes in *P. mume* and *P. mume* var. *tortuosa* and other species. The protein sequences of 20 *At*MPKs, 17 *Os*MPKs, 10 *At*MKKs, and 8 *Os*MKKs were retrieved from the TAIR 10 (http://www.arabidopsis.org/, accessed on 5 October 2022) and TIGR (http://rice.plantbiology.msu.edu/, accessed on 5 October 2022) databases, respectively. The *P. mume* genome sequence and annotation files were downloaded from the *P. mume* genome project (http://prunusmumegenome.bjfu.edu.cn/, accessed on 6 October 2022); the genomes of *P. mume* var. *tortuosa* [50] and five other Rosaceae species, including *Prunus armeniaca* [52], *Prunus persica* [53], *Prunus salicina* [54], *Malus domestica* [55] and *Rosa chinensis* [56] were downloaded from the Genome Database for Rosaceae (https://www. rosaceae.org/, accessed on 6 October 2022).

### 4.2. Identification of MPK and MKK Gene Family Members

To identify the *MPK* and *MKK* genes in *P. mume* and *P. mume* var. *tortuosa*, we conducted a BLASTP against 2 *P. mume* and *P. mume* var. *tortuosa* genomes and 5 Rosaceae species with MPK and MKK protein sequences of Arabidopsis as queries with an E-value threshold set at 10^−10^. The SMART database (http://smart.embl-heidelberg.de/, accessed on 10 October 2022) and the Conserved Domain Database of NCBI (https://www.ncbi.nlm.nih.gov/cdd, accessed on 10 October 2022) were then used to confirm all putative MPK and MKK proteins.

The *MPK* and *MKK* genes were named based on their similarity to Arabidopsis MPK and MKK proteins. Additionally, the number of amino acids, molecular weight (MW), and isoelectric point (pI) were computed using the Python script. The subcellular localization of *MPK* and *MKK* gene family members was evaluated using the Cell-PLoc 2.0 online server [71] (http://www.csbio.sjtu.edu.cn/bioinf/Cell-PLoc-2/, accessed on 20 October 2022).

### 4.3. Phylogenetic Analysis

In order to investigate the phylogenetic relationships between *MPK* and *MKK* genes in *P. mume* and *P. mume* var. *tortuosa* and other species, the alignment of full-length MPK and MKK protein sequences from *P. mume*, *P. mume* var. *tortuosa*, *A. thaliana*, and *O. sativa* along with five other Rosaceae species was carried out using MAFFT software with the auto setting [72]. Subsequently, phylogenetic trees with maximum likelihood (ML) were created using FastTree (version 2.1.11) [73] with default parameters. The phylogenetic tree was then annotated and beautified using EvolView v2 [74] (https://www.evolgenius.info/evolview-v2, accessed on 24 October 2022).

### 4.4. Conserved Motif, Conserved Domain, and Gene Structure of MPK and MKK Proteins in *P. mume* and *P. mume* var. tortuosa

MEME Suite version 5.3.3 (https://meme-suite.org/meme/tools/meme, accessed on 8 November 2022) [75] predicted the conserved motifs of each identified *Pm*MPK, *Pmv*MPK, *Pm*MKK, and *Pmv*MKK protein. The number of motifs for the conserved domains was set to 13, the motif width was set to 6–50 amino acids, and the residuals were set to the default parameters. The *P. mume* and *P. mume* var. *tortuosa* genome gff files were used to extract the gene structure data, which were then visualized with TBtools software [76] and edited in AI CS6.

### 4.5. Chromosome Location, Duplication, and Synteny Analysis

Gene location and genome chromosome length information for each *PmMPK*, *PmMKK*, *PmvMPK*, and *PmvMKK* gene of *P. mume* and *P. mume* var. *tortuosa* were retrieved from the gff file downloaded from the *P. mume* genome project (http://prunusmumegenome.bjfu.edu.cn/, accessed on 6 October 2022) and the Genome Database for Rosaceae (https://www. rosaceae.org/, accessed on 6 October 2022), respectively. Using TBtools, a chromosomal location figure was created. The Multiple Collinearity Scan Toolkit (MCScanX) and Advanced Circos in TBtools were used to analyze gene tandem and segment replication events. The criteria used for identifying gene duplication were as follows: (a) all genes are initially classified as ‘singletons’ and assigned gene ranks according to their order of appearance along chromosomes; (b) in any BLASTP hit, the two genes are re-labeled as ‘tandem duplicates’ if they have a difference of gene rank = 1, that is, they were continuous repeat; (c) the anchor genes in collinear blocks are re-labeled as ‘segment replications’ [76,77]. The synteny analysis of the *MPK* and *MKK* genes of *P. mume* and *P. mume* var. *tortuosa*, respectively, with *A. thaliana* and five Rosaceae species was performed using MCScanX in TBtools. The rate of Ka (non-synonymous substitution)/Ks (synonymous substitution) was employed to assess the codon evolutionary rate between the duplicated gene pairs based on the alignment of the coding sequence using the Ka/Ks calculator program embedded in the TBtools. According to 2 ordinary rates (λ) of 1.5 × 10^−8^ and 6.1 × 10^−9^ substitutions per site per year [78,79], the formula t = Ks/2λ × 10^−6^ Mya was used to calculate the divergence time.

### 4.6. Cis-Acting Element Analysis of *P. mume* and *P. mume* var. tortuosa MPK and MKK Gene Promoter Regions

The upstream genomic sequences (2.0 kb) of each identified *PmMPK*, *PmMKK*, *PmvMPK*, and *PmvMKK* gene were extracted from the genomic sequence data using TBtools and then uploaded to the PlantCARE database (http://bioinformatics.psb.ugent.be/webtools/plantcare/html/, accessed on 22 November 2022) [80] for cis-acting element analysis. After much deliberation, we selected 14 elements for *MPK* and 13 for *MKK*, including some that are activated by hormones such as methyl jasmonate (MeJA) and abscisic acid (ABA). Additionally, some elements respond to growth and development (such as light response and meristem expression) as well as stress, such as low temperatures. The map was created by TBtools using these data along with phylogenetic tree data (nwk file) and then edited by AI CS6 software.

### 4.7. PmMPK and PmMKK Gene Expression Analysis

We examined the expression patterns of *PmMPKs* and *PmMKKs* in various tissues using data from RNA sequencing of the root, stem, leaf, flower bud, and fruit of wild *P. mume* collected from Tongmai, Tibet, China (30°06′ N, 95° 05′ E) (RNAseq data are available in the NCBI Gene Expression Omnibus (GEO) under accessions GSE40162) [46], and we examined the responses of *PmMPKs* and *PmMKKs* to naturally low temperatures from November to February using data from RNA sequencing of *P. mume* ‘Lve’ flower bud dormancy, the specimens for which were grown in the Jiufeng International Plum Blossom Garden, Beijing, China (40°07′ N, 116°11′ E) (the data was acquired from the corresponding author) [47]. Additionally, we investigated the expression of *MPK* and *MKK* gene family members of *P. mume* ‘Songchun’ for three seasons, including autumn (cold acclimation, October), winter (the final period of endo-dormancy, January), and spring (deacclimation, March) in three geographical locations: Beijing (BJ, 39°54′ N, 116°28′ E), Chifeng (CF, 42°17′ N, 118°58′ E), and Gongzhuling (ZGL, 43°42′ N, 124°47′ E) to better understand the role of *PmMPKs* and *PmMKKs* in reacting to natural low temperatures (the data was acquired from the author). Heatmaps were created using TBtools [76].

### 4.8. qRT–PCR Analysis of PmMPK and PmMKK Genes

To validate the results of transcriptome analysis of *P. mume MPK* and *MKK* genes during naturally low-temperature, we collected the annual branches of the cold-tolerant *P. mume* ‘Songchun’ and the cold-sensitive cultivar *P. mume* ‘Lve’ from the Jiufeng International Plum Blossom Garden, Beijing, China (40°07′ N, 116°11′ E) to study how *PmMPKs* and *PmMKKs* react to low temperatures (Figure 16). Before chilling treatment, the shoots were incubated at 22 °C for an overnight period; then, they were lowered to 4 °C under long-day circumstances (16 h of light and 8 h of darkness) for 0, 1, 4, 6, 12, 24, 48, and 72 h. The stems were immediately removed and put in liquid nitrogen, then kept at −80 °C for long-term storage in preparation for RNA extraction. Each treatment had three biological replicates.

The RNAprep Pure Plant Plus Kit (Tiangen, Beijing, China) was used to extract the total RNA from each sample. Using ReverTra Ace® qPCR RT master mix with gDNA remover (Toyobo, Osaka, Japan), complementary cDNA was synthesized. Primer 3 (https://bioinfo.ut.ee/primer3–0.4.0/, accessed on 2 December 2022) designed the specific primers based on the cDNA sequences (Appendix A). Quantitative real-time polymerase chain reaction (qRT–PCR) was undertaken on a qTOWER2.2 system (Analytik Jena, Jena, German) with a SYBR® Green Premix Pro Taq HS qPCR kit (Accurate Biology, Hunan, China) to examine the expression levels of *PmMPKs* and *PmMKKs* at low temperatures. The reactions took place at a 20 μL volume and contained 4.0 μL 10 × each of forward and reverse primers, 2.0 μL 10× of cDNA, and 10.0 μL SYBR® Green Premix Pro Taq HS qPCR master mix. The reactions were carried out using the following protocol: 95 °C for 30 s, followed by 40 cycles of 95 °C for 5 s, 55 °C for 30 s, and 72 °C for 30 s. The annealing temperature was adjusted according to the actual situation; specifically, temperatures were 55 °C for *PmMPK4/5/6/7/13*; 57 °C for *PmMPK1* and *PmMKK3;* 58 °C for *PmMPK3/16/19/20* and *PmMKK9-1/2/6*; and 60.6 °C for *PmMPK8*. Using the *PHOSPHATASE 2A* (*PP2A*) and *Actin* gene of *P. mume* as the reference gene, the relative expression was calculated using the delta–delta CT method [81]. For each qRT–PCR, three biological replicates were used. Using SPSS 22.0, separate statistical analyses of ‘Songchun’ and ‘Lve’ were performed. With a significant difference at level *p* = 0.05, the Student–Newman–Keuls test and least significant difference (LSD) test were used to calculate the one-way ANOVA analysis of variance. The graphs were produced using GraphPad Prism6 software.

## 5. Conclusions

In conclusion, our research is the first to carry out the genome-wide identification and characterization of *MPKs* and *MKKs* in the species *P. mume* and its variety *P. mume* var. *tortuosa*, including chromosomal location, the identification of duplication genes, analysis of gene structure, phylogenetic relationships, and conserved motifs. Based on the RNA-seq data, the expression profiles of the *PmMPK* and *PmMKK* genes in various tissues and geographical locations were also further examined. Furthermore, under cold stress conditions, a qRT–PCR assay was used to examine the expression profiles of the *PmMPK* and *PmMKK* genes. Our findings may be very important for further research into the biological roles of *PmMPKs* and *PmMKKs*.

## Figures and Tables

**Figure 1 ijms-24-08829-f001:**
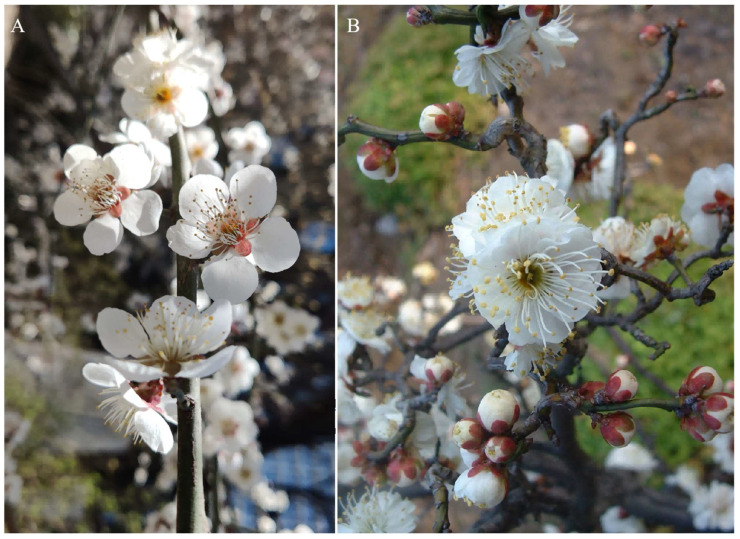
The image of the wild *P. mume* (**A**) and *P. mume* var. *tortuosa* (**B**).

**Figure 2 ijms-24-08829-f002:**
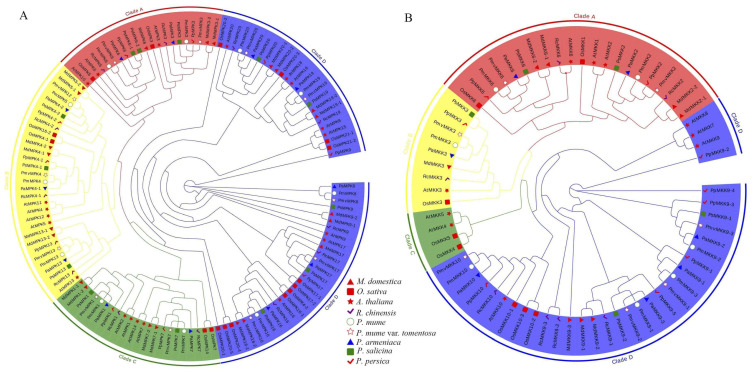
Phylogenetic tree of *MPK* (**A**) and *MKK* (**B**) sequences from *P. mume*, *P. mume* var. *tortuosa*, and other plant species. Clades A, B, C, and D are indicated by red, yellow, green, and blue branch lines, respectively. At, *A. thaliana*; Os, *O. sativa*; Pa, *P. armeniaca*; Pm, *P. mume*; Pmv, *P. mume* var. *tortuosa*; Pp, *P. persica*; Ps, *P. salicina*; Md, *M. domestica*; Rc, *R. chinensis*. Different species are indicated with shapes and colors.

**Figure 3 ijms-24-08829-f003:**
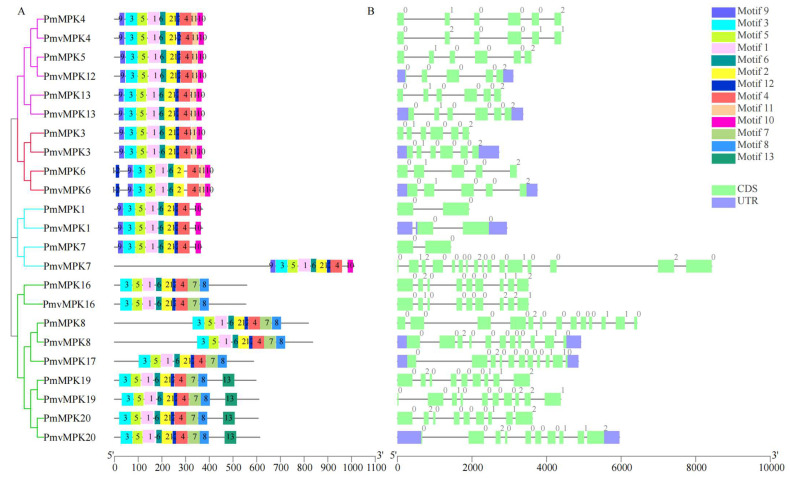
Phylogenetic relationship, conserved motifs, and gene structures analysis of *PmMPK* and *PmvMPK* genes. (**A**) The ML phylogenetic tree and motif composition of *PmMPK* and *PmvMPK* genes. The *MPK* genes are grouped into four clades, A (red), B (magenta), C (cyan), and D (green). Thirteen motifs are shown in different colored rectangles. (**B**) Exon-intron organization of *P. mume* and *P. mume* var. *tortuosa MPK* genes. Spring green and black correspond to exons and introns, respectively, and light slate blue corresponds to UTR.

**Figure 4 ijms-24-08829-f004:**
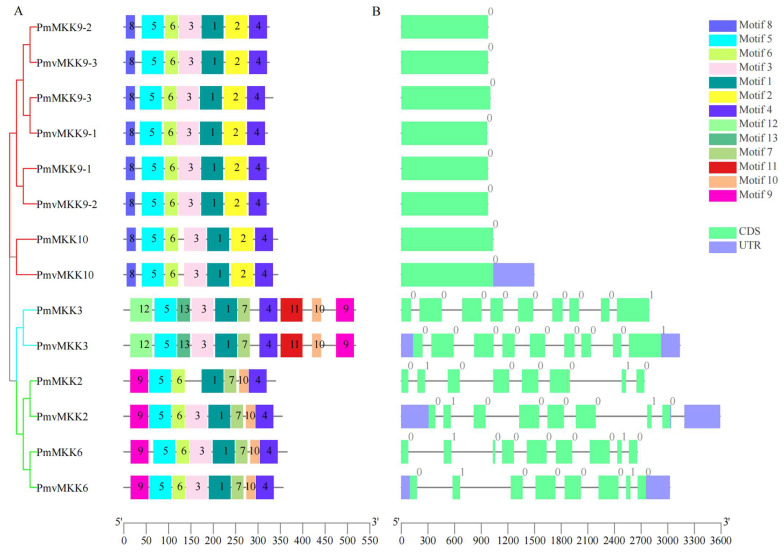
Phylogenetic relationship, conserved motifs, and gene structures analysis of *PmMKK* and *PmvMKK* genes. (**A**) The ML phylogenetic tree and motif composition of *PmMKK* and *PmvMKK* genes. The *MKK* genes of *P. mume* and *P. mume* var. *tortuosa* are grouped into three clades, A (green), B (cyan), D (red). Thirteen motifs are shown in different colored rectangles. (**B**) Exon-intron organization of *P. mume* and *P. mume* var. *tortuosa MKK* genes. Spring green and black correspond to exons and introns, respectively, and light slate blue corresponds to UTR.

**Figure 5 ijms-24-08829-f005:**
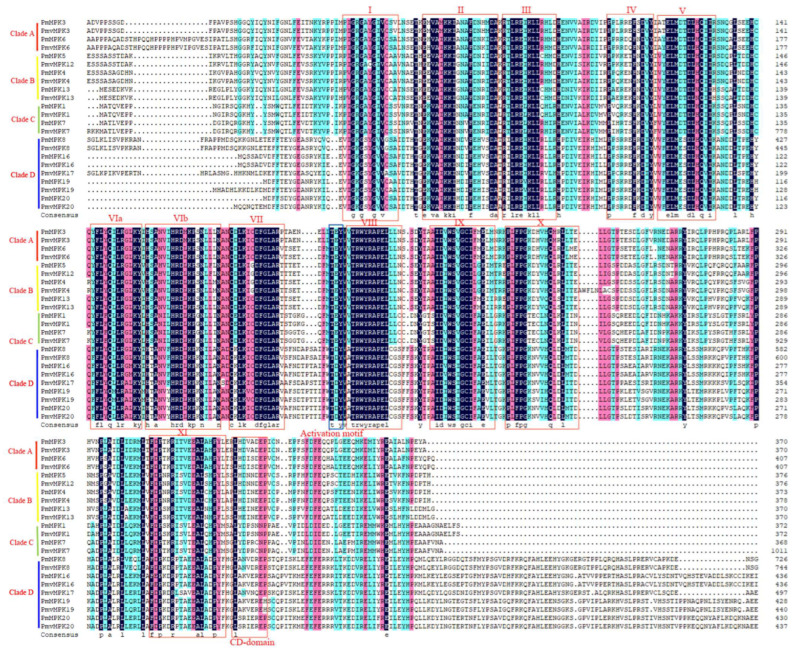
Sequence alignment and motif analysis of *MPK* family in *P. mume* and *P. mume* var. tortuosa. Key motifs of -TEY- in Clade A, B, and C and -TDY- in Group D within the activation loop are marked by a blue rectangle; CD-domain is marked by a red rectangle. The 11 conserved domains are in roman numerals (I to XI) above the sequence with a red rectangle.

**Figure 6 ijms-24-08829-f006:**
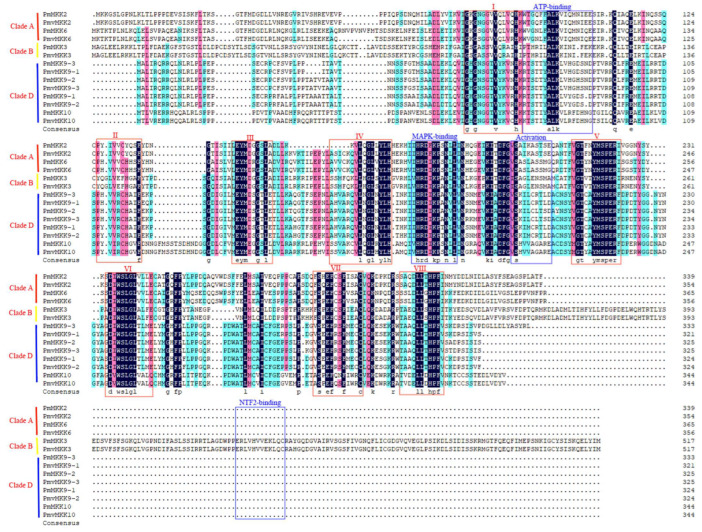
Sequence alignment and motif analysis of *MKK* family in *P. mume* and *P. mume* var. tortuosa. The conserved domains (I to VIII, ATP-binding, MAPK-binding, Activation, and NTF2-binding domain) present in protein kinase are denoted by roman numerals with red and blue rectangles.

**Figure 7 ijms-24-08829-f007:**
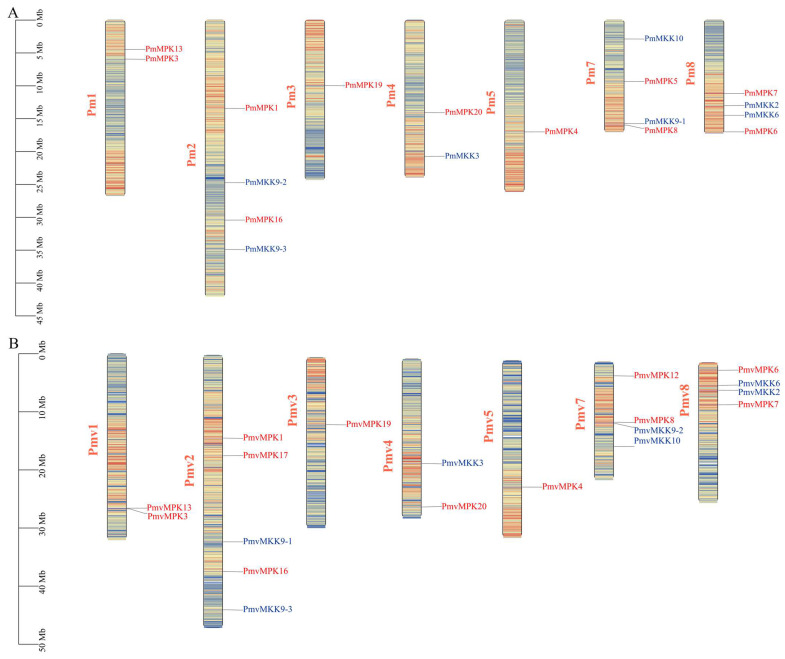
Schematic representations of the chromosomal location of the *MPK* and *MKK* genes in *P. mume* (**A**) and *P. mume* var. *tortuosa* (**B**). The chromosome number is indicated on the left of each chromosome. Pm, *P. mume*; Pmv, *P. mume* var. *tortuosa*.

**Figure 8 ijms-24-08829-f008:**
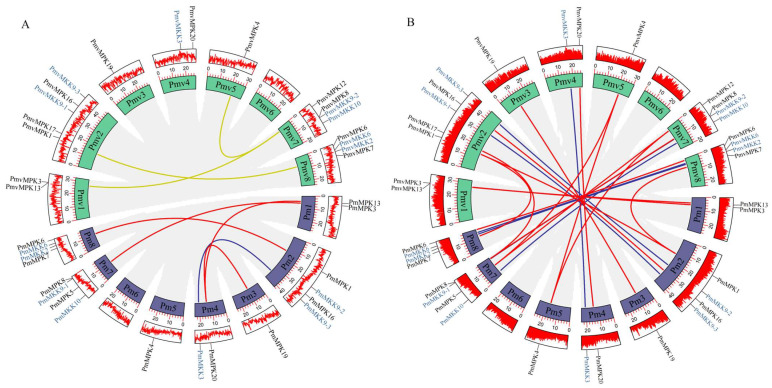
The Circos figure for the *MPK* and *MKK* genes in *P. mume* and *P. mume* var. *tortuosa* segmental duplication links and collinearity analysis. (**A**) *PmMPK*, *PmMKK*, and *PmvMPK* segmental duplication links. The red, blue, and olive-drab lines indicate *PmMPK*, *PmMKK*, and *PmvMPK* segmented duplicated gene pairs, respectively. (**B**) Collinearity analysis of *MPK* and *MKK* genes in *P. mume* and *P. mume* var. *tortuosa*. The red and blue lines represent *MPK* and *MKK* collinearity gene pairs, respectively. The red line in the outermost circle represents the gene density distribution.

**Figure 9 ijms-24-08829-f009:**
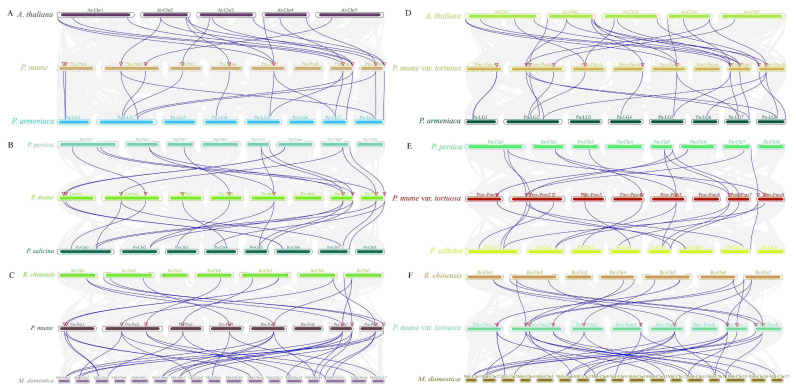
Collinearity analysis of MPK genes among *P. mume*, *P. mume* var. *tortuosa*, *A. thaliana*, *P. armeniaca*, *P. persica*, *P. salicina*, *M. domestica*, and *R. chinensis* genomes. (**A**) *A. thaliana* (At) vs. *P. mume* (Pm) vs. *P. armeniaca* (Pa). (**B**) *P. persica* (Pp) vs. *P. mume* (Pm) vs. *P. salicina* (Ps). (**C**) *R. chinensis* (Rc) vs. *P. mume* (Pm) vs. *M. domestica* (Md). (**D**) *A. thaliana* (At) vs. *P. mume* var. *tortuosa* (Pmv) vs. *P. armeniaca* (Pa). (**E**) *P. persica* (Pp) vs. *P. mume* var. *tortuosa* (Pmv) vs. *P. salicina* (Ps). (**F**) *R. chinensis* (Rc) vs. *P. mume* var. *tortuosa* (Pmv) vs. *M. domestica* (Md). Colored circular rectangles denote the chromosomes of three plants. Grey curves indicate collinear blocks within the genomes, and the blue curves represent collinear gene pairs with MPK genes. The red triangle represents the location of the PmMPK and PmvMPK genes.

**Figure 10 ijms-24-08829-f010:**
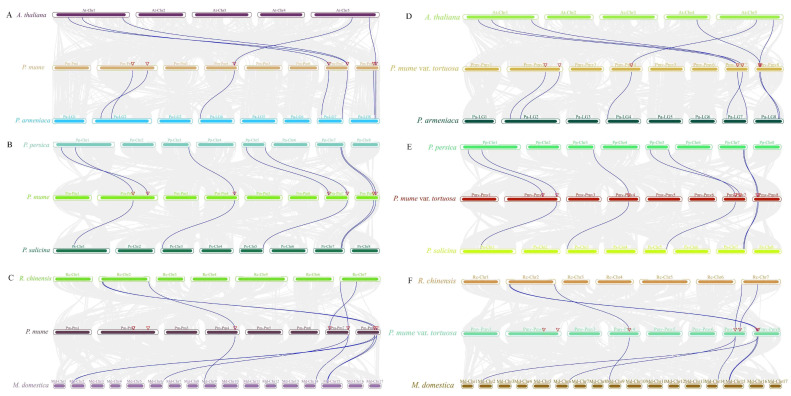
Collinearity analysis of *MKK* genes among *P. mume, *P. mume** var. *tortuosa*, A*. thaliana, P. armeniaca, P. persica, P. salicina, M. domestica*, and *R. chinensis* genomes. (**A**) *A. thaliana* (At) vs. *P. mume* (Pm) vs*. P. armeniaca* (Pa). (**B**) *P. persica* (Pp) vs. *P. mume* (Pm) vs*. P. salicina* (Ps). (**C**) R*. chinensis* (Rc) vs*. *P. mume** (Pm) vs *M. domestica* (Md). (**D**) *A. thaliana* (At) vs*. *P. mume** var. *tortuosa* (Pmv) vs. *P. armeniaca* (Pa). (**E**) *P. persica* (Pp) vs*. *P. mume** var. *tortuosa* (Pmv) vs. *P. salicina* (Ps). (**F**) *R. chinensis* (Rc) vs*. *P. mume** var. *tortuosa* (Pmv) vs. *M. domestica* (Md). Colored circular rectangles denote the chromosomes of three plants. Grey curves indicate collinear blocks within the genomes, and the blue curves represent gene pairs that are collinear with *MKK* genes. The red triangle represents the location of the *PmMKK* and *PmvMKK* genes.

**Figure 11 ijms-24-08829-f011:**
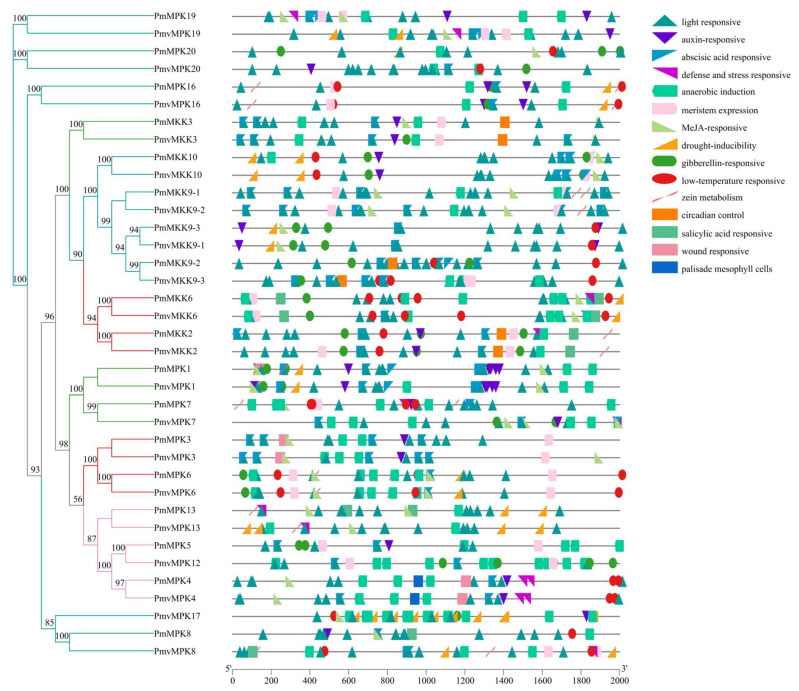
Predicted cis-elements in *PmMPK*, *PmvMPK*, *PmMKK*, and *PmvMKK* gene promoters that react to hormone response, stress response, and plant growth regulation. The distribution of the main 14 cis-elements in *PmMPK* and *PmvMPK*, 13 cis-elements in *PmMKK* and *PmvMKK* gene promoters. Different color shapes in patterns indicate various elements and their locations in each *PmMPK*, *PmvMPK*, *PmMKK*, and *PmvMKK* promoter. The *MPK* and *MKK* genes are classified into four clades, and red, Magenta, green, and Cyan represent Clades A, B, C, and D, respectively.

**Figure 12 ijms-24-08829-f012:**
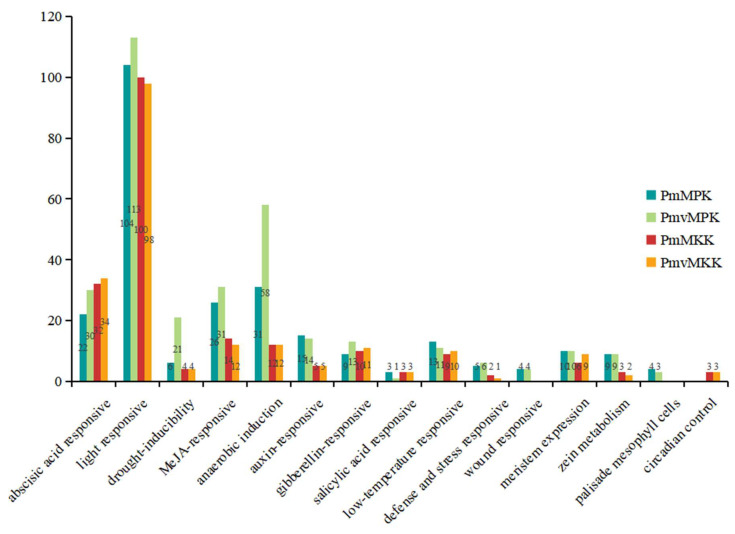
The number of cis-elements in *PmMPK*, *PmvMPK*, *PmMKK*, and *PmvMKK* promoter.

**Figure 13 ijms-24-08829-f013:**
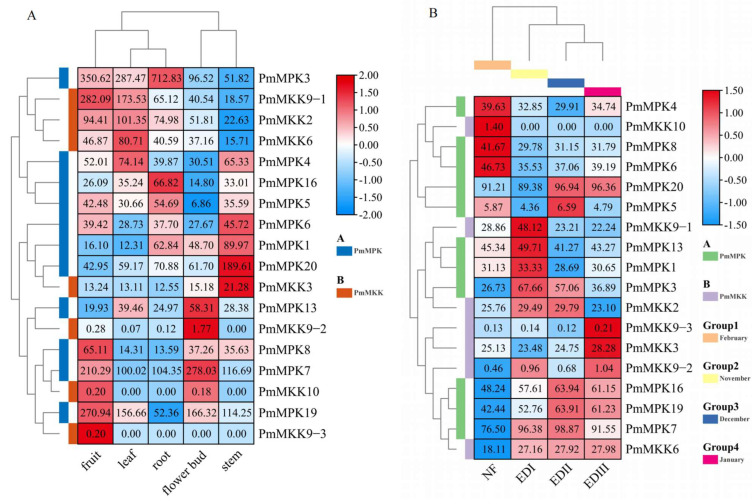
*PmMPK* and *PmMKK* gene expression profiles in various tissues and flower bud stages. (**A**) *PmMPKs* and *PmMKKs* expression patterns in various tissues. (**B**) *PmMPKs* and *PmMKKs* expression patterns in flower buds during dormancy. EDI: Endo-dormancy I, November; EDII: Endo-dormancy II, December; EDIII: Endo-dormancy III, January; NF: Natural flush, February. The expression amount is converted to a 2-based log function and then normalized by the row using the normalized method. The relative expression level is indicated by the color scale to the right of the heat map, and an elevated expression level is indicated by the color gradient from dodger blue to red.

**Figure 14 ijms-24-08829-f014:**
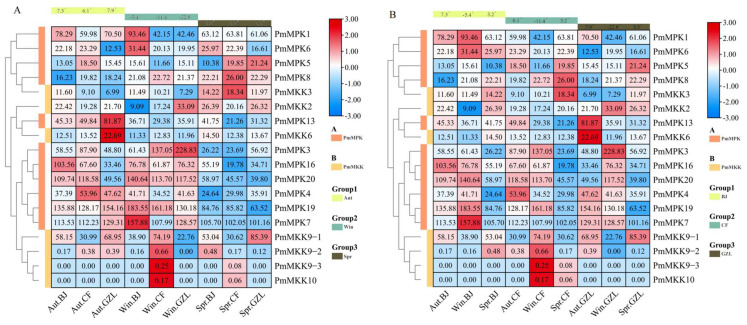
Expression profiles of *PmMPKs* and *PmMKKs* across seasons and regions. (**A**) *PmMPKs* and *PmMKKs* expression profiles of ‘Songchun’ in Beijing, Chifeng, and Gongzhuling and during various seasons (autumn, winter, and spring). (**B**) Comparison of differential expression profiles of *PmMPKs* and *PmMKKs* in Beijing, Chifeng, and Gongzhuling during different seasons. The expression amount is converted using a 2-based log function and then normalized by the row scale method. The relative expression level is indicated by the color scale to the right of the heat map, and an elevated expression level is indicated by the color gradient from dodger blue to red. Aut, Autumn; Win, Winter; Spr, Spring. BJ, Beijing; CF, Chifeng; GZL, Gongzhuling.

**Figure 15 ijms-24-08829-f015:**
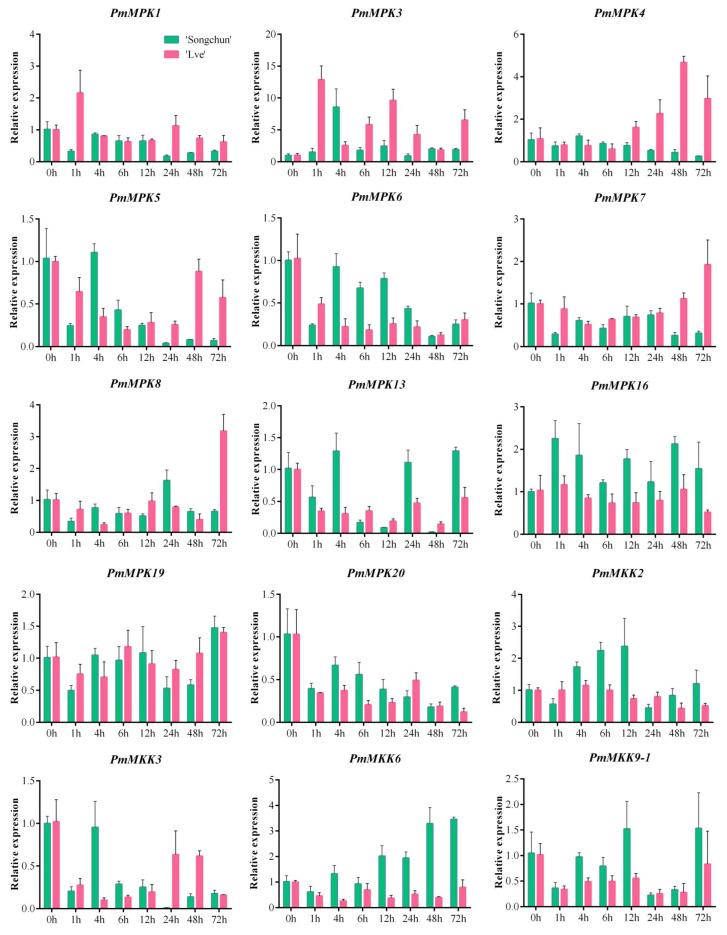
Expression patterns of 11 *PmMPK* and *4 PmMKK* genes under low-temperature treatments. The transcript levels of 11 *PmMPK* and 4 *PmMKK* genes were assessed using the relative quantification method (ΔΔCt) with the *PmActin* gene as the reference gene. The standard deviation of three biological replicates is represented by error bars. The statistical analyses of ‘Songchun’ and ‘Lve’ were conducted independently using SPSS 22.0, the one-way ANOVA analysis of variance was calculated using the least significant difference (LSD) and Student-Newman-Keuls test, difffferent letters above the bars indicate significant difffferences (*p* = 0.05). Spring green letters indicate ‘Songchun’, deep pink letters indicate ‘Lve’. The diagram was created using the GraphPad Prism6 software.

**Figure 16 ijms-24-08829-f016:**
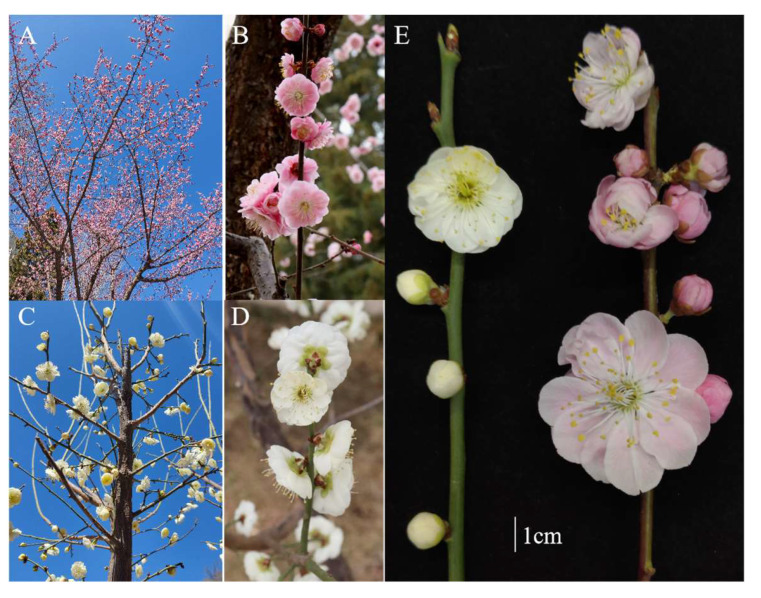
Pictures of *P. mume* ‘Songchun’ and *P. mume* ‘Lve’. (**A**,**B**) *P. mume* ‘Songchun’. (**C**,**D**) *P. mume* ‘Lve’. (**E**) *P. mume* ‘Lve’ (left) and *P. mume* ‘Songchun’ (right).

**Table 1 ijms-24-08829-t001:** Summary of *MPK* and *MKK* genes detected in *P. mume* and *P. mume* var. *tortuosa*.

Species/ Gene Family	Name	Gene ID	Clade	CDS (bp)	No. of Amino Acids	Molecular Weight (kDa)	pI	Locus	Subcellular Location
*P. mume*	*PmMPK1*	Pm005869	C	1141	372	42.71	6.47	Pm2:13426673:13428592	Nucleus
*Q.*									
MAPK	*PmMPK3*	Pm000966	A	1135	370	42.61	5.62	Pm1:5929065:5930985	Nucleus
	*PmMPK4*	Pm018234	B	1144	373	42.91	6.08	Pm5:17007404:17011789	Nucleus
	*PmMPK5*	Pm023935	B	1153	376	43.11	6.01	Pm7:9355221:9358805	Nucleus
	*PmMPK6*	Pm027774	A	1248	407	46.44	5.8	Pm8:16998106:17001298	Nucleus
	*PmMPK7*	Pm026678	C	1129	368	42.39	8.46	Pm8:11162132:11163561	Nucleus
	*PmMPK8*	Pm025094	D	2506	818	92.92	8.32	Pm7:15979218:15985644	Nucleus
	*PmMPK13*	Pm000736	B	1135	370	42.58	5.18	Pm1:4461784:4464553	Nucleus
	*PmMPK16*	Pm008036	D	1710	558	63.28	8.45	Pm2:30424089:30427599	Nucleus
	*PmMPK19*	Pm011269	D	1829	597	67.75	9.31	Pm3:9959653:9963198	Nucleus
	*PmMPK20*	Pm014593	D	1857	606	69.09	9.11	Pm4:14063001:14066620	Nucleus
*P. mume* var. *tortuosa*	*PmvMPK1*	PmuVar_Chr2_1968	C	1141	372	42.71	6.47	Chr2:14262930:14264849	Nucleus
	*PmvMPK3*	PmuVar_Chr1_3496	A	1135	370	42.61	5.62	Chr1:26706111:26708031	Nucleus
MAPK	*PmvMPK4*	PmuVar_Chr5_2162	B	1159	378	43.56	6.08	Chr5:21784736:21789120	Nucleus
	*PmvMPK6*	PmuVar_Chr8_0208	A	1248	407	46.42	5.8	Chr8:1295418:1298610	Nucleus
	*PmvMPK7*	PmuVar_Chr8_1256	C	3096	1011	115.17	8.99	Chr8:7245293:7253723	Cell membrane
	*PmvMPK8*	PmuVar_Chr7_1503	D	2561	836	94.75	8.51	Chr7:10372791:10379229	Nucleus
	*PmvMPK12*	PmuVar_Chr7_0255	B	1153	376	43.13	6.21	Chr7:2387026:2390606	Nucleus
	*PmvMPK13*	PmuVar_Chr1_3483	B	1135	370	42.58	5.18	Chr1:26582862:26585626	Nucleus
	*PmvMPK16*	PmuVar_Chr2_5103	D	1698	554	62.88	8.56	Chr2:37239461:37242972	Nucleus
	*PmvMPK17*	PmuVar_Chr2_2455	D	1799	587	66.31	6.86	Chr2:17306082:17310386	Nucleus
	*PmvMPK19*	PmuVar_Chr3_1670	D	1866	609	69.19	9.29	Chr3:11546774:11551157	Nucleus
	*PmvMPK20*	PmuVar_Chr4_3258	D	1878	613	69.95	9.06	Chr4:25471101:25474741	Nucleus
*P. mume*	*PmMKK2*	Pm027015	A	1040	339	37.84	5.36	Pm8:13011712:13014449	Nucleus
MAPKK	*PmMKK3*	Pm015648	B	1588	518	57.79	5.53	Pm4:20729768:20732555	Nucleus
	*PmMKK6*	Pm027289	A	1119	365	40.96	5.69	Pm8:14478577:14481234	Nucleus
	*PmMKK9-1*	Pm025044	D	994	324	36.17	7.58	Pm7:15725552:15726526	Nucleus
	*PmMKK9-2*	Pm007435	D	997	325	36.21	8.04	Pm2:24735502:24736479	Nucleus
	*PmMKK9-3*	Pm008654	D	1022	333	37.36	7.12	Pm2:34893630:34894631	Nucleus
	*PmMKK10*	Pm023176	D	1055	344	38.42	5.9	Pm7:2872242:2873276	Nucleus
*P. mume* var. *tortuosa*	*PmvMKK2*	PmuVar_Chr8_0833	A	1086	354	39.50	5.51	Chr8:4732675:4735395	Nucleus
	*PmvMKK3*	PmuVar_Chr4_2126	B	1588	518	57.77	5.51	Chr4:17988174:17990962	Nucleus
MAPKK	*PmvMKK6*	PmuVar_Chr8_0700	A	1092	356	39.90	5.59	Chr8:3908415:3911074	Nucleus
	*PmvMKK9*−*1*	PmuVar_Chr2_4435	D	985	321	35.95	7.12	Chr2:32122931:32123896	Nucleus
	*PmvMKK9*−*2*	PmuVar_Chr7_1550	D	994	324	36.17	7.58	Chr7:10632062:10633036	Nucleus
	*PmvMKK9*−*3*	PmuVar_Chr2_5667	D	997	325	36.08	7.14	Chr2:43816380:43817357	Nucleus
	*PmvMKK10*	PmuVar_Chr7_1991	D	1055	344	38.41	5.86	Chr7:14554221:14555255	Nucleus

## Data Availability

The data are included in Appendix A.

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
