# Peer review of "Genome-Wide Identification of the *MAPK* and *MAPKK* Gene Families in Response to Cold Stress in *Prunus mume"

_ijms, 2023, doi:10.3390/ijms24108829_

Round 1
Reviewer 1 Report
It is very meaningful for identifying these two gene families of MAPK and MAPKK in Plum Blossom. While the writing still needs a lot of improvement. You have to ask a native English-speaking colleague or use a paid editing service to check your manuscript firstly. And many Figures and the legends are arranged out of order. In addition, there are several concerns that you have to explain. Please see the followings.
"Mei" is just a generic synonym for this kind of plant in Chinese, and I don't think it is scientific for using this name everywhere, especially in English articles, so I recommend you use more scientific names for what you want to refer to.
It's not clear to me why you use these two kinds of plants to identify the relevant gene families, and what is the connection between them? I don't see a particularly interesting explanation in the article, so I suggest that they can be seperated in different articles, otherwise the article seems to be unfocused.
The varieties of the two kinds of plants for genome sequencing or genome data? The citeria of MPK and MKK? Did you check the PFs? Or only according to the homologous blast?
You have to provide the criteria of segmental and tandem duplications. I don't think you understand this enough. The Ka/Ks analysis is also not specific.
You must also describe the origin of the materials you used and the detailed information, including the name of the variety, the accession number, the original provider, etc.
Why did you only conduct the gene expressions of PmMPK and PmMKK? How about PmvMPK and PmvMKK? Are there differences in gene sequences of PmMPK and PmMKK between the varieties used in gene expression studies and those for genome sequencing or the variety with the genome data?
Reviewer 2 Report
An in-depth and complete study with important results for gene function study of MAPK and MAPKK gene families in response to cold stress in Prunus mume. More carefully proofread of the manuscript should be used in the format and statement modification. There are some minor comments.
1. line 18-19, does this sentence mean MKKs is one of the members of MPKs, or MPKs and MKKs are the members of MPKs? which should be clear on the sentence expressing.
2. Almost figures are not clear. I could not open the attachment of Figure S1-S9 in the supplementary, so please the figures in the supplementary would also be checked.
3. Almost figures are complicated. Such as in figure 2, the motif composition is clamped in the ML tree. The title of figure 5 is on the figure and the text of the results.
4. In figure 3, which genes belong to clade A, and which genes belong to clade B? I suggest the authors could supply this information in figure 3, although this information is provided in the previous results. Which would help the readers read the article with ease.
5. line 267, 15 homologous MPK genes pairs in ..., I could not find the red lines in Figure 7.
6. The section 2.5, the interspecies collinearity analysis of the MPK and MKK gene family. 18 and 19 MPKs and MKKs in P. mume and P. mume var. tortuosa. Why there are over 20 homologous gene pairs are deteted? (the level of genomes?)
7. One reference gene is chosen to calculate the relative expression of genes. I suggest multiple reference genes (double internal reference genes) would ensure the accuracy of the results of expression analysis.
Reviewer 3 Report
The manuscript (ijms-2209574) submitted by Wen et al. demonstrates the genome-wide identification of MAPK and MAPKK gene families in Prunus mume using bioinformatics approaches in combination with qRT-PCR verification. They present sufficient data about the fundamental features of the two MAPK gene families. The presented information is useful for others in future studies. However, the manuscript is not acceptable for publication in its current form. The major concerns I have are listed below:
1. The English language needs to be polished. There are many typos and/or non-scientific expressions throughout the manuscript;
2. Currently the names of MAPKs and MKKs were named based on their sequential locations on chromosomes. I think it’s better to name them based on their similarity to corresponding Arabidopsis genes or rice for better comparison and less confusion in the future;
3. In Figure 1, Figure S1, S2, different colored shapes need to be explained in the figure legends;
4. Are there any significant differences of cis-elements in MAPKs and MKKs between Prunus mume and other species?
5. Do you have expression profiles of MAPKs and MKKs genes in the cold-sensitive cultivar ‘Zaolve’ during the flower bud dormancy period?
Reviewer 4 Report
The submitted Manuscript is devoted to the interesting subject on the role of protein kinases in cold tolerance of Prunus mume.
Before considering further on this submitted Manuscripts needs to have figures of the reasonable quality, the present figures are of the quality that they are not seen with required details. There could be several rounds of revisions, alternatively the rejection because of the improper presentation of the results.
Please, provide the figure legends below the figures, the submitted Manuscript cannot be evaluated due to the low quality of its initial presentation.
More specific questions at a first glance.
1) Quantitative real-time polymerase chain reaction (qRT–PCR) was 687
used on a qTOWER2.2 System (analytikjena, Jena, German) with a SYBR Green Premix 688
Please, provide the complete name of the producer.
2) The annealing temperature is adjusted according to the actual 694
situation. Using the PHOSPHATASE 2A (PP2A) gene of P. mume as the reference gene, 695
the relative expression was calculated using the delta-delta CT method [73].
Please, indicate why the PHOSPHATASE 2A (PP2A) gene was taken and how its expression changed over time.
Were the other genes taken as the reference genes such as actin?
3) The annealing temperature is adjusted according to the actual 694
situation. Using the PHOSPHATASE 2A (PP2A) gene of P. mume as the reference gene, 695
the relative expression was calculated using the delta-delta CT method [73].
Please, indicate the annealing temperatures for each situation.
4) they have gradually formed a complex and delicate signal transduction mecha- 41
nism [1, 2].
There are many mechanisms. Please, use plural form and add more relevant references.
5) Table 1. PmvMPK12 PmuVar_Chr8_125 C 3096 1011 115.17 8.99 Chr8:7245293:7253723 Cell mem-
Please, indicate the location of the gene.
6) Figure 2 is not readable at all.
Round 2
Reviewer 1 Report
You have done a lot of work, but only briefly listed, did not compare the differences between the P. mume and the P. mume var. tortuosa in detail, and did not focus on these differences in the gene expression analysis later, just based on the gene expression differences of the two cultivars can not serve the topic of the manuscript, lack of logic. It is recommended to rewrite and resubmit.
Actually, the P. mume here refers to the wild species, while the P. mume var. tortuosa refers to domesticated cultivar, which you need to explain at the beginning and distinguish below. They are two different plants, and the manuscript should focus on comparisons between wild species and domesticated cultivar of P. mume, without involving too much analysis of other plants.
You still do not give the criteria of segmental and tandem duplications, for example, in barley we would say this,"Genes were considered to be segmentally duplicated if they occurred in collinear segments containing at least five collinear gene pairs, whereas they were considered to be tandemly duplicated if they were located close to each other with no more than two interval genes".
Reviewer 3 Report
In the revised version of the manuscript, the authors have addressed all my concerns and the revise manuscript is significantly improved.
Author Response
Thank you for your time and effort given the constructive comments to improve our manuscript entitled “Genome-wide identification of the MAPK and MAPKK gene families in response to cold stress in Prunus mume”.
Thank you very much for your consideration.
Yours sincerely,
Lidan Sun, PhD
Professor of Forest Genetics
Beijing Forestry University
Reviewer 4 Report
Nice and interesting paper sufficient for IJMS with some corrections missing yet. However, it is suffering from too strong bioinformatic approach taken. More specific questions.
1) have similar origins and evolutionary processes in P. mume and its varieties. A cis-acting
28 regulatory element analysis shows that MPK and MKK genes may function in P. mume and its
29 varieties’
One variety was studied, should be variety not varieties. Please, correct where appropriate.
2) have similar origins and evolutionary processes in P. mume and its varieties. A cis-acting
28 regulatory element analysis shows that MPK and MKK genes
and involved similar evolutionary processes
3) A cis-acting
28 regulatory element analysis shows
The role of cis-acting element have to be confirmed by extra methods further on at least.
4) abstract
Would be good to add about the tissue specific expression to the abstract from figure 13.
5) figure 6 is still not impressive without good resolution of the proteins, it’s rather for figure while the fine details are the most important for functioning. The very low resolution is given.
6) figure 6 what is “lue indicates high quality, whereas red indicates low
257 quality.” What does the quality mean in the text here?
7) It’s would be good to do biochemical assays with the proteins to check their activities.
8) Would be good to give a picture of the plant with a ruler demonstrating the size of the plant.
9) It would be good to discuss the very interesting results from figure 15 about the DIFFERENCES in shoots of the two varieties between the varieties.
So far the whole paper is too bioinformatics and mechanistic.
Round 3
Reviewer 1 Report
As I know, the P. mume and P. mume var. tortuosa are two different species here, and they represent the wild P. mume and cultivar P. mume, respectively. So it is not suitable say "...its variety...". You have revised some, but not all, please check carefully. As you say, you want to compare the differences of MPK and MKK genes between the P. mume and P. mume var. tortuosa. I think you should further compare the differences in the expression of these genes between them, including the response to cold treatment. I don't understand why you compare these genes between two different cultivars with different cold tolerance. It is a different story.